# Prediction of Early Season Nitrogen Uptake in Maize Using High-Resolution Aerial Hyperspectral Imagery

**Tyler J. Nigon** [1,2], **Ce Yang** [2,*], **Gabriel Dias Paiao** [1], **David J. Mulla** [1], **Joseph F. Knight** [3] **and Fabián G. Fernández** [1]

[1] Department of Soil, Water and Climate, University of Minnesota, Saint Paul, MN 55108, USA; nigo0024@umn.edu (T.J.N.); gdiaspai@umn.edu (G.D.P.); mulla003@umn.edu (D.J.M.); fabiangf@umn.edu (F.G.F.)
[2] Department of Bioproducts and Biosystems Engineering, University of Minnesota, Saint Paul, MN 55108, USA
[3] Department of Forest Resources, University of Minnesota, Saint Paul, MN 55108, USA; jknight@umn.edu
[*] Correspondence: ceyang@umn.edu; Tel.: +1-612-626-6419

**Abstract:** The ability to predict spatially explicit nitrogen uptake (NUP) in maize (*Zea mays* L.) during the early development stages provides clear value for making in-season nitrogen fertilizer applications that can improve NUP efficiency and reduce the risk of nitrogen loss to the environment. Aerial hyperspectral imaging is an attractive agronomic research tool for its ability to capture spectral data over relatively large areas, enabling its use for predicting NUP at the field scale. The overarching goal of this work was to use supervised learning regression algorithms—Lasso, support vector regression (SVR), random forest, and partial least squares regression (PLSR)—to predict early season (i.e., V6–V14) maize NUP at three experimental sites in Minnesota using high-resolution hyperspectral imagery. In addition to the spectral features offered by hyperspectral imaging, the 10th percentile Modified Chlorophyll Absorption Ratio Index Improved (MCARI2) was made available to the learning models as an auxiliary feature to assess its ability to improve NUP prediction accuracy. The trained models demonstrated robustness by maintaining satisfactory prediction accuracy across locations, pixel sizes, development stages, and a broad range of NUP values (4.8 to 182 kg ha$^{-1}$). Using the four most informative spectral features in addition to the auxiliary feature, the mean absolute error (MAE) of Lasso, SVR, and PLSR models (9.4, 9.7, and 9.5 kg ha$^{-1}$, respectively) was lower than that of random forest (11.2 kg ha$^{-1}$). The relative MAE for the Lasso, SVR, PLSR, and random forest models was 16.5%, 17.0%, 16.6%, and 19.6%, respectively. The inclusion of the auxiliary feature not only improved overall prediction accuracy by 1.6 kg ha$^{-1}$ (14%) across all models, but it also reduced the number of input features required to reach optimal performance. The variance of predicted NUP increased as the measured NUP increased (MAE of the Lasso model increased from 4.0 to 12.1 kg ha$^{-1}$ for measured NUP less than 25 kg ha$^{-1}$ and greater than 100 kg ha$^{-1}$, respectively). The most influential spectral features were oftentimes adjacent to each other (i.e., within approximately 6 nm), indicating the importance of both spectral precision and derivative spectra around key wavelengths for explaining NUP. Finally, several challenges and opportunities are discussed regarding the use of these results in the context of improving nitrogen fertilizer management.

**Keywords:** cross-validation; feature selection; hyperparameter tuning; image processing; image segmentation; nitrogen fertilizer recommendation; supervised regression

## 1. Introduction

Nitrogen (N) fertilizer inputs are crucial for achieving high crop yields, but the loss of reactive N from agricultural systems leads to atmospheric, surface water, and groundwater pollution [1–3],

ultimately diminishing environmental quality and human well-being [4,5]. Despite the potential environmental consequences to society, the pressure on producers to increase productivity oftentimes leads to N fertilizer applications in excess of crop requirement [6,7]. Without more efficient N fertilizer applications, the increasing global population and subsequent rising demand for food are expected to cause an increase in the loss of reactive N in the future [8].

A strategy to reduce the likelihood of reactive N loss is to apply part of the crop's N requirement after emergence, delaying the application until crop demand is near its maximum. Grain yields and crop N use are not uniform across seasons [9], so this delayed application also provides the opportunity to adapt N fertilizer rates in a dynamic manner based on the influence of weather and N cycle processes on early season crop growth or stress. Maize N requirement typically varies spatially [10], so this strategy can be more effective with variable rate/site-specific applications [11].

The fundamental barrier to our ability of making reliable site-specific N fertilizer recommendations is the lack of understanding and inability to accurately quantify soil N supply and how it is expected to change throughout the season [12]. This is particularly challenging because both natural N supply (i.e., non-fertilizer) and crop N requirement are dynamic in both space and time and are difficult to predict [13]. Determination of early season crop N uptake (NUP) can be helpful for making N fertilizer recommendations due to its connection with natural N supply (e.g., mineralization) and crop N requirement. Remote sensing offers the opportunity to capture near real-time information about crop N status [14], and it can be an efficient way to assess the spatial variability across an entire field or farm. With the availability of robust, reliable unmanned aircraft and low payload hyperspectral line-scanning imagers in recent years, there is now the opportunity to use high-resolution, aerial hyperspectral imagery to predict early season NUP in maize. As a research tool, aerial hyperspectral imaging is attractive for its ability to scale, especially compared to other methods that use ground-based hyperspectral point measurements for N fertilizer management [15].

There are tradeoffs in pixel size related to the approach for which remote sensing imagery is captured. It is typically preferred to capture images with finer spatial resolution, but this comes with a tradeoff of larger data storage and processing requirements, as well as longer acquisition time and battery/fuel requirement for a given spatial extent. Academic research is commonly conducted on small plot areas that do not cover a large extent, and a low flight altitude (e.g., approximately 20 m) may be practical for covering the extent of the experimental area. However, this is rarely suitable at the field scale (including on-farm research) because of the large areas that must be imaged. This leads to a likely dilemma where much of the data used for research and development is inherently different from the data used by practitioners (e.g., the inherent pixel size differs). Therefore, it is important to develop prediction models that are either scale-agnostic or that are specifically tailored to the conditions for which practitioners will be able to practically implement them.

The objectives of this work were to: (i) compare supervised learning techniques for their ability to predict maize NUP at the early- to mid-vegetative development stages (i.e., V6 to V14) using spectral features from high-resolution aerial hyperspectral imagery, (ii) quantify the potential model improvement by including an auxiliary feature derived before the segmentation process, and (iii) evaluate how the difference in pixel size at image capture affects prediction accuracy based on the inherent ability to segment pixels most influenced by soil and/or shadow.

## 2. Materials and Methods

### 2.1. Field Experiments

Data from three experiments in southern Minnesota were used to evaluate the objectives of this study. The Wells experiment (43.85437, –93.72977) was conducted near Wells, Minnesota in 2018, and the two Waseca experiments were conducted at the Agroecology Research Farm (44.063635, –93.540281) near Waseca, Minnesota in 2019 (hereby referred to as the Waseca "small plot" and "whole field" experiments). Weed, pest, and disease management were carried out by farm managers using approaches typical

for the areas. The N fertilizer rate across all experiments ranged from 0 to 231 kg ha$^{-1}$. Experimental treatments included conventional and sidedress N applications, whereby the conventional treatments had all N applied around the time of planting, and the sidedress treatments had 45 kg ha$^{-1}$ N applied at planting and the remainder applied as sidedress close to the V8 development stage.

### 2.1.1. Wells Experiment

The Wells, Minnesota dataset was acquired from a broader experiment that evaluated the effect of tillage (conventional tillage, strip-till, and no-till) and tile drainage (drained and undrained) on N response and contained 144 experimental plots across four replications. The previous crop (i.e., in 2017) was soybeans, and the Marna silty clay loam (fine, smectitic, mesic Vertic Endoaquolls) and Nicollet silty clay loam (fine-loamy, mixed, superactive, mesic Aquic Hapludolls) soils were the two predominant soil series at the site. The crop was planted (Pioneer hybrid P9929AMXT) on 17 May 2018 at a population of 86,400 plants ha$^{-1}$. Nitrogen was applied as urea + N-(n-butyl) thiophosphoric triamide (NBPT; urease inhibitor) on 21 May 2018. Phosphorus and potassium fertilizers were applied on 25 May 2018 at rates that ensure optimum soil fertility for maize production in Minnesota [16].

### 2.1.2. Waseca Experiments

The experimental boundaries for both the small plot and whole field experiments near Waseca, Minnesota are illustrated in Figure 1 [17]. The previous crop (i.e., in 2018) was soybeans, and the Nicollet clay loam (fine-loamy, mixed, superactive, mesic Aquic Hapludolls) and Webster clay loam (fine-loamy, mixed, superactive, mesic Typic Endoaquolls) soils were the two predominant soil series at the site. Both the small plot and whole field experiments were planted (Channel hybrid 199-1VT2P RIB) on 3 June 2019 at a population of 87,722 plants ha$^{-1}$ immediately following a tillage pass. Preplant N was applied to the small-plot experiment as urea + NBPT on 17 May 2019. The preplant N treatment was applied to the whole field experiment in two phases: the first phase was applied as a urea + NBPT/ammonium sulfate blend (45 kg N ha$^{-1}$) on 16 May 2019, and the second phase was applied as urea + NBPT at a rate of 179 kg nitrogen ha$^{-1}$ on 3 June 2019. Sidedress N was applied as urea + NBPT on 3 July 2019 (at the V7 development stage) with a 4.6 m applicator (Gandy Orbit-Air, Gandy Company; Owatonna, Minnesota; application width of eight crop rows) equipped with a variable rate controller (Viper version 3.2.0.0, Raven Industries, Inc.; Sioux Falls, South Dakota).

### *2.2. Crop Nitrogen Uptake*

To convert biomass from a per-plant basis to a per-area basis, a plant emergence of 83,361 plants ha$^{-1}$ was assumed for the Wells experiment (96.4% emergence), and plant emergence of 85,251 plants ha$^{-1}$ was assumed for both Waseca experiments (97.0% emergence). These emergence assumptions were based on average stand count observations conducted shortly after emergence. Within two days of capturing hyperspectral imagery, tissue samples were collected to measure tissue N concentration in the above-ground biomass and above-ground total NUP. Plants were sampled by cutting the base of the stem just above the soil surface. Additional details regarding sample acquisition (i.e., date, development stage, number of samples, etc.) are listed in Table A1. Plant samples were oven-dried at 60 °C to constant weight, weighed, and ground to pass a 1 mm sieve. Total N concentration was determined for each sample using Kjeldahl digestion [18] (Wells experiment) or dry combustion [19] (Waseca experiments).

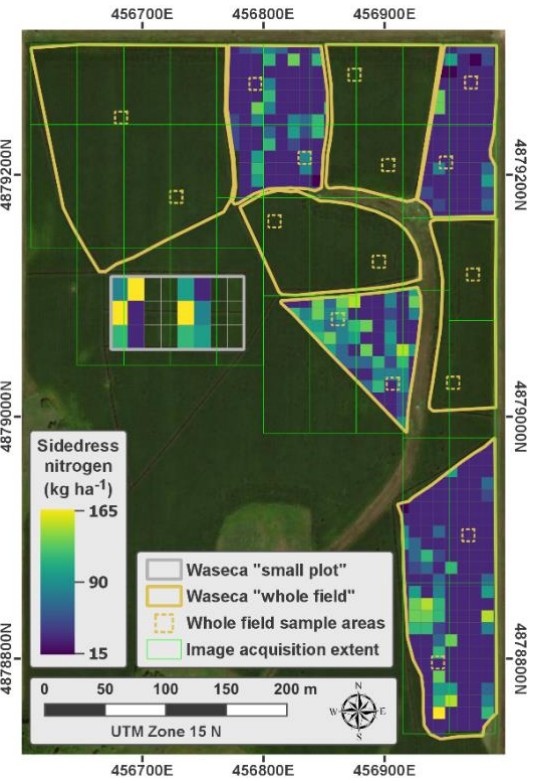

**Figure 1.** Waseca "small plot" and "whole field" experimental boundaries. The whole field sample areas, whole-field hyperspectral image extents, and sidedress N fertilizer rates are also illustrated.

*2.3. Airborne Spectral Imaging System*

Hyperspectral images were captured via an airborne spectral imaging system (Resonon, Inc., Bozeman, Montana). The airborne system included a flight computer (Resonon NUMI, flight code version 1.36) that integrated the following hardware for seamless data capture: (i) a pushbroom (i.e., line scanning) hyperspectral imaging spectrometer (Resonon Pika IIg VNIR), (ii) an inertial measurement unit (IMU; Ellipse2-N, SBG Systems, Carrières-sur-Seine, France), (iii) a single band GNSS (Global Navigation Satellite System) antenna (TW2410, Tallysman, Inc., Ottawa, Ontario) surface mounted and lifted 15 cm away from the unmanned aerial vehicle flight controller, and (iv) a 240 GB solid-state hard drive for data storage (Intel S3500 Series). The specifications of the spectral imager are presented in Table 1. The flight computer and spectral imager were powered by a designated lithium polymer battery (Ronin 4350 mAh, DJI, Inc., Shenzhen, China), and they were rigidly mounted to a three-axis gimbal (Ronin-MX, DJI, Inc., Shenzhen, China) on board an unmanned aerial vehicle (DJI Matrice 600 Pro, DJI, Inc., Shenzhen, China).

**Table 1.** Specifications of the hyperspectral imaging spectrometer.

| Spectral Range (nm) | Spectral Resolution (nm) | Spectral Channels | Spatial Channels | Bit Depth | Field of View (Degrees) |
|---|---|---|---|---|---|
| 400–900 | 2.1 | 240 [1] | 640 | 12 | 33.0 |

[1] Although 240 spectral channels were available, many were clipped out because of high noise.

*2.4. Airborne Image Capture*

Resonon Ground Station software (version 3.123) was used to program the flight computer before each flight campaign. The "bright areas of interest" auto-exposure setting was used, which adjusted the gain and exposure time based on ambient lighting conditions and the general brightness of the target when each image began to be captured. A moderate framerate of 109 (frames per second) was used for

each image campaign because it was suitable for the desired ground speeds and altitudes for achieving spatial integrity (i.e., pixels captured with a 1:1 aspect ratio in the cross-track and along-track directions).

A polygon that delineated the desired area for image capture, together with real-time location data from the GNSS receiver of the airborne system, dictated whether the spectral imager was to capture data at a particular location. Image capture commenced when the airborne system entered the polygon boundary and ceased when it exited the polygon boundary.

The gimbal was programmed to maintain the position of the spectral imager at nadir, which involved real-time gimbal adjustments (pitch and roll), due mostly to UAV movement and wind force. The gimbal maintained the heading of the spectral imager relative to the heading of the UAV so that the spectral image array (640 spatial channels) remained approximately perpendicular to the heading of the UAV flight direction. Autonomous flight missions were created and executed using DJI Ground Station Pro (iOS app version 1.8.2 + for 2018 flights and iOS app version 2.0.2 + for 2019 flights). The altitude, speed, and resulting image pixel size (i.e., ground-resolved distance) for each experimental site are presented in Table 2.

**Table 2.** Spatial description of aerial image campaigns at each experimental site. The cropped plot size refers to the spatial extent of cropped image at each sample location.

| Site | Altitude | Ground Speed | Ground Swath | Pixel Size | Area Captured | Cropped Plot Size |
|------|----------|--------------|--------------|------------|---------------|-------------------|
| | m | $\text{m s}^{-1}$ | m | cm | ha | m |
| Wells | 40 | 4.0 | 23.7 | 4.0 | 4.5 | $6.2 \times 1.8$ |
| Waseca small-plot [1] | 20–25 | 2.0–2.5 | 11.8–14.8 | 2.0–2.5 | 0.7 | $1.8 \times 1.8$ |
| Waseca whole-field | 80 | 8.0 | 47.4 | 8.0 | 11.2 | $10 \times 10$ |

[1] Altitude was 20 m on the V6 and V8 development stages, but it was 25 m at the V14 stage to increase the ground swath. The corresponding cropped plot size at V14 was 1.8 m × 2.3 m.

### 2.5. Reference Panels

Reference panels were constructed by applying a 50/50 mixture of gray paint and barium sulfate ($BaSO_4$) by weight to high-density fiberboard (60 cm × 60 cm × 3.2 mm). Painted $BaSO_4$ diffuses incoming solar irradiance in various directions to minimize specular reflection and has less than a 5% difference from a Lambertian surface for 0 to 55 degree zenith angles [20]. The spectral profile of the reference panels from approximately 400 to 900 nm was nearly flat, with an average reflectance of 42.1% ($\sigma = 1.1\%$; measured with a spectroradiometer; ASD FieldSpec 4; Analytical Spectral Devices, Inc., Longmont, Colorado). Immediately before spectral image acquisition, the reference panels were strategically placed throughout the experimental area so that images containing reference panels were captured at least every 90 s to account for temporal variation in solar illumination.

### 2.6. Image Pre-Processing

All image pre-processing steps were carried out using Spectronon software (version 2.134; Resonon, Inc., Bozeman, Montana). Raw hyperspectral images were corrected to radiance ($\mu\text{W sr}^{-1} \text{ cm}^{-2} \mu\text{m}^{-1}$) in a two-step process using the Radiance Conversion plugin (Equations (1) and (2)) and gain and offset files supplied by the manufacturer of the spectral imager. The *Radiance Conversion plugin* first scaled the gain file (denoted as *a*) to the integration time and gain of the data cube (denoted as *x*):

$$\bar{a} = a * \left( \frac{a_\tau * a_\rho}{x_\tau * x_\rho} \right) \tag{1}$$

where $\bar{a}$ is the scaled gain file, $a_\tau$ is the integration time of the *gain* file, $a_\rho$ is the gain of the *gain* file, $x_\tau$ is the integration time of the data cube, and $x_\rho$ is the gain of the data cube. The raw data cube was then corrected to radiance:

$$X_{rad} = \left( \frac{X_{dn}}{\bar{a}} - b \right) * R \tag{2}$$

where $X_{rad}$ is the radiance corrected data cube in units of $\mu W \, sr^{-1} \, cm^{-2} \, \mu m^{-1}$, $X_{dn}$ is the raw data cube, $b$ is the offset file with the closet match to $x_\tau$ and $x_\rho$, and $R$ is the calibrated radiance source that was used to illuminate the integrating sphere.

Following radiometric correction, images were georectified (using the *Georectification plugin*) based on time-synced data collected by the GNSS receiver and IMU of the airborne system (i.e., latitude, longitude, altitude, yaw, pitch, and roll). A 1.0 m digital elevation model image was used as the basis for projecting each image line to the appropriate elevation above mean sea level. The line-scanning imager sometimes fails to capture data when it passes over the target area too quickly, and it conversely captures duplicate data when it passes too slowly. The *Georectification plugin* used a linear interpolation method to adjust pixel values where there was missing or duplicate data.

Then, georectified radiance images were converted to reflectance by applying a single spectrum correction to each image [21] based on the relationship between the radiance of the reference panels from the imagery and the average measured reflectance. Radiance pixels were manually extracted, taking care to avoid pixels near the edges of panels. If there was no reference panel present in an image, radiance data extracted from the reference panel captured closest in time were used for reflectance conversion instead.

### 2.7. Image Post-Processing

All image post-processing was performed in Python (version 3.7.3) using the *hs_process* package (version 0.0.3) [22]; the *hs_process* package leverages *Spectral Python* (version 0.20) [23] and the *Geospatial Data Abstraction Library* (version 2.3.3) [24]. Post-processing steps included cropping, spectral clipping and smoothing, and segmentation.

#### 2.7.1. Cropping

To allow for seamless batch processing on all subsequent processing and analysis steps, spectral images were spatially subset to exclude all areas outside the plot boundary by cropping the images to plot boundaries. The image size following the spatial cropping step for each experimental location is summarized in Table 2.

#### 2.7.2. Spectral Clipping and Smoothing

Since there was low signal-to-noise, the spectral bands shorter than 430 nm ($n = 13$) and longer than 880 nm ($n = 3$) were clipped (i.e., removed) from every image cube. The spectral bands near the $O_2$ absorption region [25] (760–776 nm; $n = 7$) and $H_2O$ absorption region [26] (813–827 nm; $n = 7$) were also clipped because of relatively high noise. In total, 210 spectral bands were kept for analysis. Following spectral clipping, the Savitzky–Golay smoothing procedure (11-band moving window, second-order polynomial fitting) was applied to the spectral domain of each image pixel [27]. Following these steps, at least one cropped spectral image existed for every plot used for analysis.

#### 2.7.3. Choice of the Auxiliary Feature and Image Segmentation

The MCARI2 (Modified Chlorophyll Absorption Ratio Index Improved) spectral index (Equation (3)) was applied to each image to segment the vegetation pixels from those that represent soil and shadow. MCARI2 was chosen because it incorporates a soil adjustment factor (based on the concept developed by Huete [28]) that was optimized with the constraint of preserving the sensitivity of MCARI2 to leaf area index and insensitivity to chlorophyll influence [29].

$$MCARI2 = \frac{1.5[2.5(\lambda_{800} - \lambda_{670}) - 1.3(\lambda_{800} - \lambda_{550})]}{\sqrt{(2 * \lambda_{800} + 1)^2 - \left(6 * \lambda_{800} - 5 * \sqrt{\lambda_{670}}\right) - 0.5}} \tag{3}$$

Before segmentation was carried out, a preliminary analysis was conducted to identify an auxiliary feature that could be made available to the learning models to accomplish the objectives of this study. Specifically, the goal of the preliminary analysis was to find a metric sensitive to above-ground biomass and/or chlorophyll concentration across the range of development stages evaluated in this study. Many pre-segmentation MCARI2 descriptive statistics (e.g., mean, median, standard deviation, various percentiles, etc.) were calculated and evaluated for their relationship with above-ground biomass and chlorophyll concentration. Following the analysis, the 10th percentile MCARI2 was the auxiliary feature chosen because of its relatively strong linear relationship with above-ground biomass across development stages.

Since the leaf area index was expected to differ across N treatments at the development stages evaluated in this study, the most appropriate MCARI2 segmentation threshold was also expected to vary. To account for these differences, the upper MCARI2 threshold was dynamically calculated as the 90th percentile within each cropped image. All pixels in the data cube with MCARI2 values below the 90th percentile threshold were masked out and excluded from any subsequent analyses. The remaining unmasked pixels were assumed to be the pixels that were most likely to represent pure vegetation and were least influenced by soil and shadow (i.e., as mixed pixels).

### 2.8. Cross-Validation

There were 247 NUP observations across four development stages (V6, V8, V10, and V14), two growing seasons (2018 and 2019), and four pixel sizes available for testing the hypotheses of this experiment (Table 2; Table A1). This full dataset was used for model training and testing, and it represented a robust combination of sampling and acquisition conditions during the early- to mid-vegetative development stages.

A stratified sampling approach was used to assign observations to the training and test sets because of the varying number of observations for each experiment and the robust combination of sampling and acquisition conditions. Stratification ensured that the training and test sets followed the same approximate probability distribution, which reduced the likelihood of model overfitting. Stratification was performed within the experiment and development stage, whereby 60% of the observations were assigned to the training set ($n = 148$) and the remaining 40% were assigned to the test set ($n = 99$). The test set was held out during training and tuning and was only used to assess the final prediction performance of the regression models. A stratified random sample from the training set (75%; $n = 111$) was used for feature selection. For hyperparameter tuning, a repeated stratified $k$-fold cross-validation technique was applied to the full training set. The cross-validation used for hyperparameter tuning implemented three repetitions and four splits (a different randomization was applied in each repetition), and hyperparameter combinations were evaluated using the mean of the 12 validation scores.

A *Yeo-Johnson* power transformation was applied to the NUP response measurements to minimize the inherent heteroscedasticity of the dataset [30].

### 2.9. Feature Selection

When limited to a fixed, small number of observations to train a model, model accuracy tends to decrease as the dimensionality of the input dataset increases. For the small number of observations in this experiment relative to the 210 original input bands, it was wise to reduce the dimensionality on the hyperspectral dataset before training the model. The aim of dimensionality reduction was to minimize multicollinearity while preserving the critical information necessary to accurately predict the response variable.

Feature selection was used as a means of dimensionality reduction and was performed before and independent of training the prediction models using the Lasso (least shrinkage and selection operator) algorithm. Lasso is a supervised regression-based model that performs regularization and identifies the most informative, least redundant features for predicting the response variable [31]. Mathematically, Lasso solves the minimization of the least-squares penalty with an added $\ell_1$ regularization term.

Feature selection is possible with Lasso because its $\ell_1$ regularization term induces sparsity by forcing the sum of the absolute value of the feature coefficients to be less than some fixed constant (controlled by the $\alpha$ parameter), effectively forcing coefficients to be exactly zero for features that are collinear and otherwise uninformative. Upon model convergence, features with coefficients equal to zero are neglected, and features with non-zero coefficients are such because they contain the richest information and are therefore allowed to contribute to the prediction model.

Lasso was implemented in Python (version 3.7.3) using the *Lasso* function of the *sklearn.linear_model* module (*scikit-learn* version 0.22) [32]. A coordinate descent algorithm was implemented to fit the feature coefficients. Since the $\alpha$ parameter controls the degree of regularization on the cost function (i.e., the penalty term) and effectively controls the number of features whose coefficients are forced to zero, $\alpha$ was adjusted from $1.0e^{+00}$ to $1.0e^{-04}$ in a logarithmic manner to explore the effect of feature number on training and cross-validation scores. The root mean squared error (RMSE) of prediction (Equation (4)) was the objective function used for scoring:

$$RMSE = \sqrt{\frac{1}{n} \sum_{i=1}^{n} (y_i - \hat{y}_i)^2} \qquad (4)$$

where $n$ is the number of observations, $y_i$ is measured NUP for the $i$-th observation, and $\hat{y}_i$ is the predicted NUP for the $i$-th observation. As $\alpha$ becomes smaller and approaches zero, the cost function becomes more similar to that of linear regression, resulting in more features having non-zero coefficients. It was common for multiple $\alpha$ values to yield a similar number of features. In such cases, the feature set with the lowest cross-validation error (i.e., the best score) was ultimately used for model testing.

### 2.10. Model Tuning and Prediction

Four supervised regression models were used to predict early season NUP: *Lasso*, *random forest*, *support vector*, and *partial least squares*. All models were fit and tuned using the *scikit-learn* Python package (version 0.22). Each of these models has hyperparameters that must be set before the learning process because they are not directly learned during model training. It was necessary to tune these hyperparameters to ensure that they were set to optimum values to avoid underfitting and to accurately test the final prediction performance of the trained model. Tuning was performed using a grid-search cross-validation technique that exhaustively searched over explicit hyperparameter ranges (implemented using the *GridSearchCV* function of the *sklearn.model_selection* module). The hyperparameter ranges evaluated for each regression model are described in the following sections. The hyperparameters that had the lowest cross-validation error (i.e., RMSE) were ultimately used for model testing.

Lasso regression is an attractive prediction model because it tends to perform well with a relatively small number of features, ultimately resulting in a simpler and more interpretable model. Lasso was tuned for its only hyperparameter, $\alpha$ (ranged from $1.0e^{+00}$ to $1.0e^{-04}$).

Support vector regression models are popular prediction models because they are typically effective in high-dimensional spaces (including those where the number of features is greater than the number of samples), they are memory efficient during training, and they are versatile in that different kernel functions can be specified as their decision function. SVR is unique in that it aims to find a function $f(x)$ that deviates from $y_i$ by no more than $\varepsilon$, while simultaneously remaining as flat as possible [33,34]. The SVR model was tuned for both the *linear* and *radial basis* kernel functions. The *linear* kernel function was tuned on the $\varepsilon$ (ranged from $1.0e^{+00}$ to $1.0e^{-03}$) and $C$ (ranged from 200 to 800) hyperparameters, and the *radial basis* kernel function was tuned on the $\varepsilon$ (ranged from $1.0e^{+00}$ to $1.0e^{-03}$), $C$ (ranged from 10 to 70), and $\gamma$ (ranged from 1 to 20) hyperparameters.

Random forests are attractive because they are invariant to scaling and transformations of feature values, they are robust to the inclusion of irrelevant features, and they produce models that can be inspected [35]. The random forest model uses an ensemble learning method in which many decision trees are learned in parallel (i.e., there is no interaction between the various trees during the learning

process) [36]. The random forest model was tuned for the *maximum number of features to consider* (ranged from $n_{features}$ to $n_{features}/20$) and the *minimum number of samples required to split an internal node* (ranged from 2 to 10).

Partial least squares regression (PLSR) transforms input features into a new feature space by forming a linear combination of features (i.e., components) that maximizes the covariance between its components and the response variable(s). As a result of this dimensionality reduction (i.e., feature extraction), PLSR can be used with any number of features [37]. It is especially useful when there are more features than observations, or when the features are highly collinear (e.g., in hyperspectral image analysis). The PLSR model was tuned for two hyperparameters: the *number of components* (ranged from 2 to 10) and whether *scaling* was implemented.

Model accuracy was evaluated using both the RMSE (Equation (4) and mean absolute error (MAE) of prediction (Equation (5))):

$$MAE = \frac{1}{n} \sum_{i=1}^{n} |y_i - \hat{y}_i|. \tag{5}$$

## 3. Results

### 3.1. Image Segmentation and MCARI2 Analysis

The MCARI2 histograms illustrate the approximate distribution of the vegetative fraction in each image before segmentation (Figure 2). The range of the MCARI2 distribution decreased as development stage progressed, and the 90th percentile MCARI2 generally increased. Images captured from the Waseca whole field experiment that had a coarser 8 cm pixel size (Figure 2d–f) showed a consistently increasing trend in the 90th percentile MCARI2 up to the V14 development stage, whereas images captured from the Waseca small-plot experiment that had a finer 2–2.5 cm pixel size (Figure 2a–c) peaked by the V8 development stage. The histograms generally shifted from a skewed right distribution at the V6 development stage to a skewed left distribution at the V14 development stage, indicating an increase in vegetative cover as the crop developed. This trend is especially apparent for the finer pixel size, providing clear evidence that the image pixel size affects the distribution of MCARI2 values.

In general, there is a higher likelihood that unmasked pixels represent pure vegetation as the 90th percentile MCARI2 increases. Imagery captured at a coarser pixel size showed a consistently lower 90th percentile MCARI2 across the mid- to late-vegetative development stages (Figure 3b). Alternatively, the larger pixel size showed a consistently higher 10th percentile MCARI2 (Figure 3a). Across all development stages, the mean difference between the smaller and larger pixel sizes was 0.12 and –0.09 for the MCARI2 10th and 90th percentiles, respectively.

The 90th percentile MCARI2 could adequately segment pure vegetation (Figure 2), but it had a weak relationship with above-ground biomass due to saturation after the V10 development stage (Figure 4b). The coefficient of determination ($R^2$) and root mean squared error (RMSE; smaller is better) improved moderately when considering only the V6, V8, and V10 development stages (the best fit from ordinary least squares regression is represented by the dashed line in Figure 4b), but it is not ideal to have a model that is constrained by development stage. However, the 10th percentile MCARI2 did not saturate out as extensively and had a strong relationship with above-ground biomass (Figure 4a; $R^2$ of 0.73 and RMSE of 907 kg ha$^{-1}$).

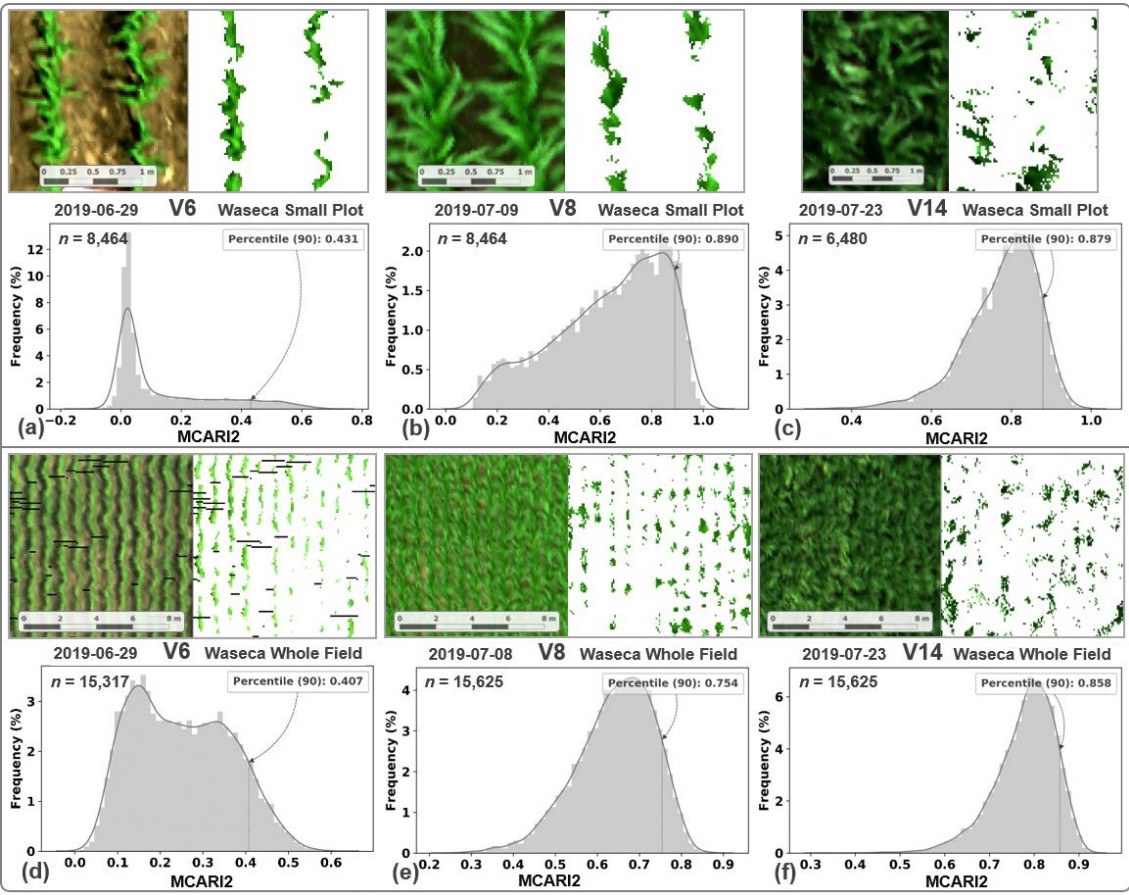

**Figure 2.** Aerial hyperspectral images (represented as a true color render) for the Waseca small plot (**a**–**c**) and Waseca whole field (**d**–**f**) experiments before and after segmentation at the V6 (a and d), V8 (b and e), and V14 (c and f) development stages. The histograms illustrate the 90th percentile MCARI2 values (Modified Chlorophyll Absorption Ratio Index Improved) for each image, which was used as the threshold for segmentation.

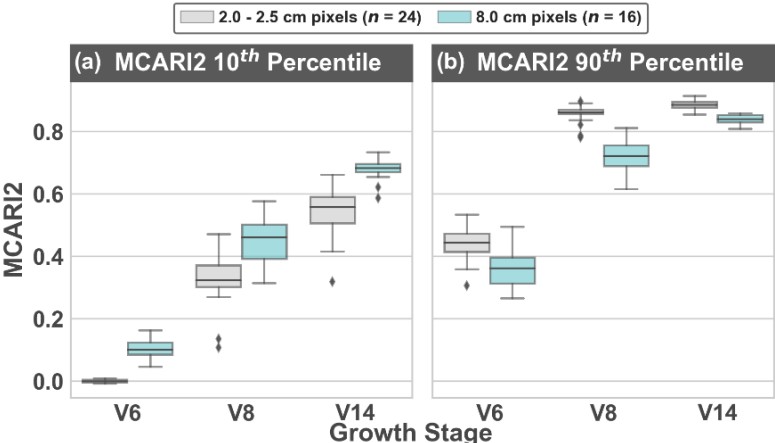

**Figure 3.** Effect of pixel size on (**a**) MCARI2 (Modified Chlorophyll Absorption Ratio Index Improved) 10th and (**b**) 90th percentile MCARI2 values at the V6, V8, and V14 development stages before image segmentation. The pixel size was 2.0 cm at the V6 and V8 development stage, but 2.5 cm at the V14 development stage.

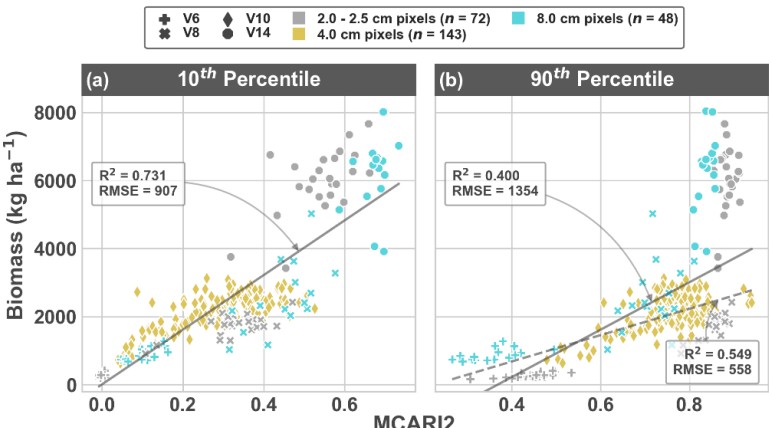

**Figure 4.** Relationship between above-ground biomass and MCARI2 (Modified Chlorophyll Absorption Ratio Index Improved) 10th (**a**) and 90th (**b**) percentile values. Best fit lines from ordinary least squares linear regression are illustrated, as well as their respective coefficient of determination ($R^2$) and root mean squared error (RMSE). The dashed line in (b) represents the best fit from ordinary least squares regression when considering only data from the V6, V8, and V10 development stages. The MCARI2 10th percentile was chosen as the auxiliary feature to complement the spectral features for model training because it did not saturate out as extensively as the MCARI2 90th percentile.

### 3.2. Feature Selection

Adjusting the $\alpha$ parameter from $1.0e^{+00}$ to $1.0e^{-04}$ in the Lasso algorithm governed the number of non-zero feature coefficients from 1 to 79, respectively (Figure 5). Few selected features were in the visible spectral region, and *all* features came from the red edge and near-infrared regions when less than 10 features were selected. The spectral areas of particular significance for NUP were from 735 to 744 nm and from 867 to 879 nm (as indicated by the features that were selected when the number of features was less than five). It is unclear why exactly adjacent features (i.e., within just a few spectral bands) were oftentimes selected by Lasso, but this may suggest that the change in spectral reflectance between adjacent bands (i.e., derivative spectra) is meaningful for explaining NUP.

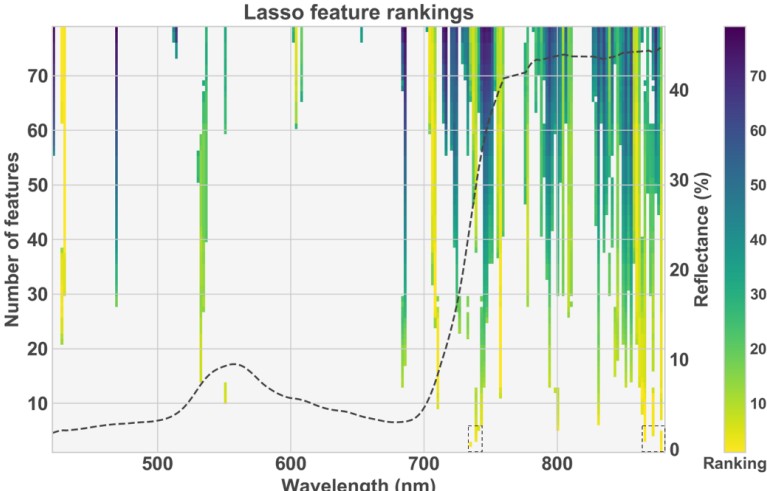

**Figure 5.** Hyperspectral features selected by the Lasso algorithm (the colored areas indicate selected features). The color indicates the ranking of the feature coefficients determined by the regularization term (yellow color corresponds to a ranking with higher feature coefficients). The top five features from 735 to 744 nm and from 867 to 879 nm are outlined to emphasize the most significant spectral regions for NUP. The mean hyperspectral reflectance across all segmented images used in this study (i.e., mean vegetation) is overlaid for reference.

*3.3. Hyperparameter Tuning*

The most common hyperparameter values across all training instances (i.e., for varying numbers of selected features) for each prediction model are listed in Table 3. The frequency of the mode indicates the proportion of training instances for which the listed hyperparameter value was optimal. Lower frequencies correspond to hyperparameters whose optimal value varied across the range of tuning values, whereas higher frequencies correspond to hyperparameters that were not particularly variable.

**Table 3.** The mode and frequency of the mode across all training instances for the tuned hyperparameters in each of the prediction models. Hyperparameter tuning results are listed for the analysis that used spectral features only, as well as the analysis that used spectral features in addition to the auxiliary feature. The hyperparameters and their values are listed according to the conventions of the *scikit-learn* function arguments [32].

| Model Parameters | Spectral Features Only | | With Auxiliary Feature | |
|---|---|---|---|---|
| | **Modal Value** | **Frequency** | **Modal Value** | **Frequency** |
| Lasso | | | | |
| alpha | 0.001 | 61% | 0.0001 | 100% |
| Support vector regression | | | | |
| kernel | "rbf" | 82% | "linear" | 92% |
| Gamma [1] | 5 | 40% | - | - |
| C [1] | 30 | 38% | 200 | 84% |
| epsilon [1] | 0.01 | 48% | 0.01 | 86% |
| Random forest | | | | |
| min_samples_split | 2 | 82% | 2 | 46% |
| max_features | 0.3 | 70% | 0.9 | 61% |
| Partial least squares | | | | |
| n_components | 7 | 39% | 7 | 61% |
| scale | 1 | 77% | 1 | 79% |

[1] The denoted support vector regression hyperparameters are summarized across only the training instances with the modal kernel function.

Lower Lasso $\alpha$ values (i.e., $1.0e^{-03}$ to $1.0e^{-04}$) were optimal, probably because lower $\alpha$ values offer less restriction on the number of selected features. In SVR, the *radial basis* kernel function was optimal when only spectral features were used, but the *linear* kernel function was optimal with the addition of the 10th percentile MCARI2 from the image segmentation step. In random forest, the *minimum number of samples required to split an internal node* was usually optimal at two, and the *maximum number of features to consider at each tree node* was typically either 30% or 90% of the total number of features. In PLSR, the optimal *number of components* generally increased as the number of features increased (*data not shown*). Although the optimal *number of components* was seven across all training instances, the number of components never exceeded five with less than 12 features (however, note that there cannot be more components than features in PLSR).

As the number selected features increased, the training and validation errors of the optimally tuned models generally decreased and the $R^2$ values increased (Figure 6). Most of the error reduction occurred in the addition of the first few features, and improvement in model fit usually plateaued by approximately 10 selected features (PLSR plateaued at approximately 30 features; data not shown). Of the four models evaluated, random forest showed more overfitting during hyperparameter tuning than Lasso, SVR, or PLSR as illustrated by the greater difference between the training and validation errors/$R^2$ values. This difference tended to increase as the number of selected features increased (although the magnitude and consistency of this observation varied among models), indicating overfitting with an increasing number of selected features. As a result of a more overfit model, the subsequent validation error in the random forest model was greater than that of Lasso, SVR, and PLSR, whose validation errors were all comparable. The standard deviation ($\sigma$) of error across the four folds and three repetitions of the repeated k-fold cross-validation (i.e., the shaded region of Figure 6) remained fairly constant

as the number of selected features increased, and it was comparable across all four models ($\sigma$ was approximately 2 kg ha$^{-1}$).

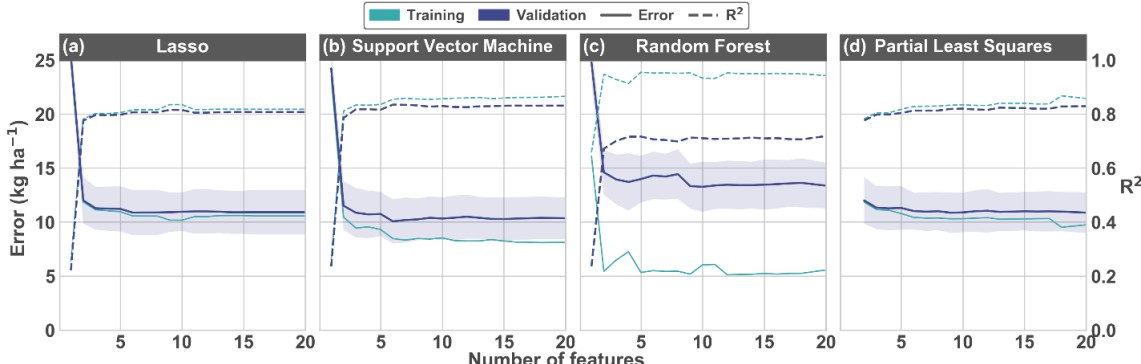

**Figure 6.** Influence of feature number on the mean absolute error (MAE) and coefficient of determination ($R^2$) during hyperparameter tuning. The MAE and $R^2$ are illustrated for both the training and validation datasets for nitrogen uptake predicted by the (**a**) Lasso, (**b**) support vector regression, (**c**) random forest, and (**d**) partial least squares models evaluated in this study. The shaded region surrounding the validation error represents the standard deviation ($\sigma$) of error across the four folds and three repetitions of the repeated k-fold cross-validation applied during hyperparameter tuning.

### 3.4. Nitrogen Uptake Predictions

Across the three experimental sites, NUP between the V6 and V14 development stages ranged from 4.8 to 181.5 kg ha$^{-1}$ (mean measured NUP in the test set was 57.1 kg ha$^{-1}$; Figure 7). There was a clear improvement in NUP prediction performance when the 10th percentile MCARI2 was available as an auxiliary feature (Figure 7e–h). All predictions in Figure 7 used five input features (predictions from Figure 7a–d used only spectral features, whereas predictions from Figure 7e–h used the 10th percentile MCARI2 auxiliary feature in addition to four spectral features). By adding the auxiliary feature, the average MAE and RMSE across the four prediction models were reduced by 1.6 (14%) and 1.8 (11%) kg ha$^{-1}$, respectively. Overall, the performance among Lasso, SVR, and PLSR was comparable, while the performance of random forest was substantially inferior as indicated by the higher error values. Among the models that used only spectral features (Figure 7a–d), the MAE of SVR was marginally lower than Lasso and PLSR (however, the RMSE was comparable). Comparing among the models that used the 10th percentile MCARI2 (Figure 7e–h), the MAE of Lasso and PLSR were marginally lower than SVR. The best-fit lines of all models are remarkably similar to the 1:1 line, especially for the models that used only spectral features, indicating that there was no tendency to under- or overpredict across the observational range of NUP. Among the three image capture pixel sizes, there does not appear to be a trend in the contribution of error by pixel size (i.e., the larger pixel size does not appear to have higher error, per se). Across all models, the relative MAE improved from 20.3% to 17.5% and the relative RMSE improved from 28.0% to 24.8% by adding the auxiliary feature (Table A2).

The *Yeo-Johnson* power transform served its purpose, as the variance of predicted NUP increased proportionally to the increase in NUP itself. This is illustrated by the increasing deviation of the points from the 1:1 line as measured NUP increased, and it is apparent by observing the increase in stratified error values shown along the right axis of each subplot. The SVR model with spectral features only performed particularly well at predicting low and very low NUP values (i.e., NUP < 50 kg ha$^{-1}$) as indicated by the low stratified MAE (i.e., 3.5 kg ha$^{-1}$) compared to the other models (average MAE of 4.6 kg ha$^{-1}$). SVR was the only learning model among the four that did not show an improvement in prediction error at the very low NUP level (i.e., NUP < 25 kg ha$^{-1}$) by making use of the 10th percentile MCARI2 feature. The SVR stratified MAE actually increased from 3.5 to 3.9 kg ha$^{-1}$ (all others decreased at the very low NUP stratification), but they were still the lowest

among the other learning models. Although the random forest models had higher overall MAE, they performed satisfactory at the very low NUP level (i.e., fairly similar to the other models). Although there were small differences in MAE within the medium and high NUP levels (NUP ≥ 50 kg ha$^{-1}$), neither Lasso, SVR, or PLS clearly emerged as the best performer. The NUP stratifications reported in Figure 7 roughly coincide with the four observed development stages. In general, the MAE increased as the development stage progressed (data not shown), which was largely attributed to the increasing variance in measured NUP with the increase in NUP itself.

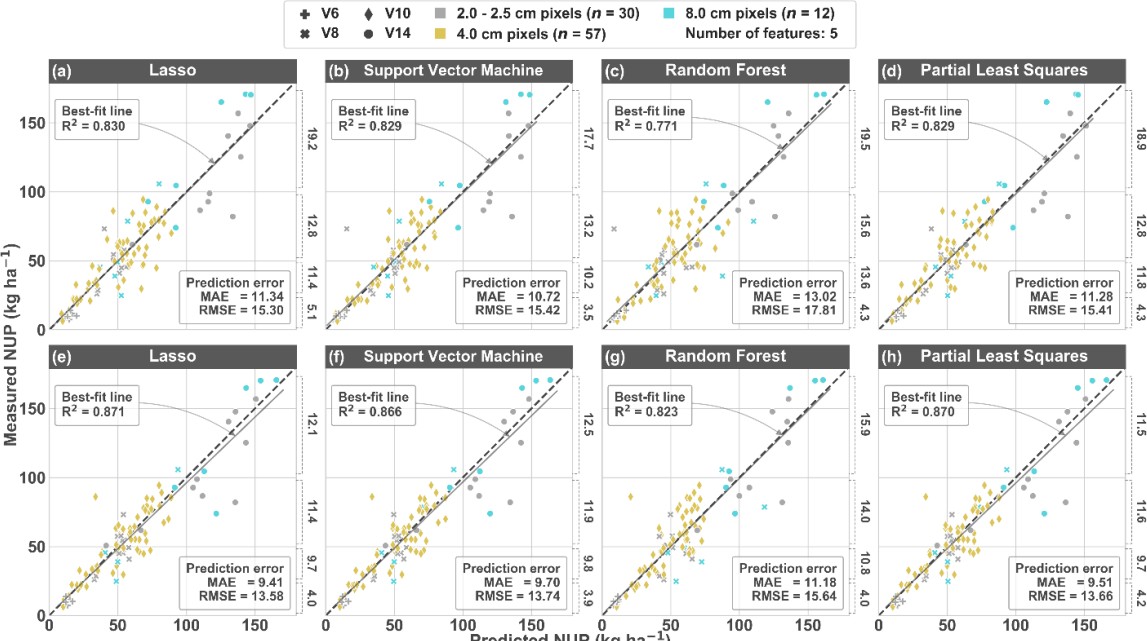

**Figure 7.** Measured and predicted nitrogen uptake (NUP) values from each of the four learning models using the test dataset (*n* = 99). Plots in the top row (**a**–**d**) used five spectral features (739, 741, 801, 867, and 873 nm), and plots in the bottom row (**e**–**h**) used four spectral features (737, 739, 867, and 871 nm) in addition to the 10th percentile MCARI2 (Modified Chlorophyll Absorption Ratio Index Improved) before image segmentation. The mean absolute error (MAE) and root mean squared error (RMSE) correspond to the prediction error (i.e., deviation from the 1:1 dashed line), whereas the coefficient of determination (R$^2$) corresponds to the best-fit line. MAE values stratified by measured NUP are shown along the right axis of each plot ("very low", "low", "medium", and "high" groups are stratified as: *NUP* < 25 kg ha$^{-1}$; 25 ≥ *NUP* < 50 kg ha$^{-1}$; 50 ≥ *NUP* < 100 kg ha$^{-1}$; and *NUP* ≥ 100 kg ha$^{-1}$, respectively).

For any number of input features, the NUP prediction performance improved when the 10th percentile MCARI2 was available as an auxiliary feature (Figure 8). Across all models, the MAE of prediction was reduced by 1.2 kg ha$^{-1}$ when considering the 10th percentile MCARI2 (summarized data not shown). Average improvements within models ranged from 0.96 kg ha$^{-1}$ (SVR model) to 1.4 kg ha$^{-1}$ (Lasso model). The number of input features required to reach optimal performance was reduced with the 10th percentile MCARI2 feature available (optimal performance is indicated by the minimum MAE across all number of input features). The Lasso, SVR, and PLSR models reached optimal performance with less than 10 input features with the 10th percentile MCARI2 auxiliary feature available. If using spectral features only, Lasso, SVR, and PLSR did not usually reach optimal performance until at least 20 input features were used. This evidence indicates that the inclusion of the 10th percentile MCARI2 value before segmentation not only improved overall prediction accuracy, but it also reduced the number of input features required to reach optimal performance (optimal prediction errors were usually reached with only three features).

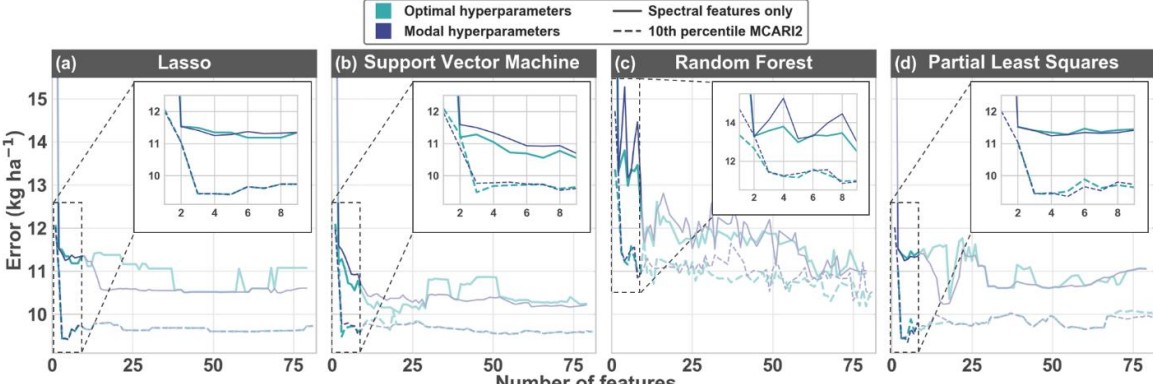

**Figure 8.** Influence of feature number on the mean absolute error (MAE) of the test dataset for Lasso (**a**), support vector regression (**b**), random forest (**c**), and partial least squares (**d**) models. The error designated by *optimal hyperparameters* indicates that the model hyperparameters were adjusted for each input feature number based on the results of the hyperparameter tuning, and the error designated by *modal hyperparameters* indicates that the hyperparameters were fixed for all number of features. Errors are presented for models that used spectral features only (solid lines), as well as well as for models that used the 10th percentile MCARI2 (Modified Chlorophyll Absorption Ratio Index Improved) auxiliary feature in addition to spectral features (dashed lines).

Unlike the other learning models, the random forest prediction error continued to decrease as the number of features increased, and it was not clear if the optimal performance was ever reached with the range of input features evaluated. Furthermore, random forest errors were noisy (i.e., they had a tendency to change abruptly as the number of features increased), and they were clearly higher than the other three models across all the features evaluated.

There was not a clear advantage in using the optimal hyperparameters compared to using the modal hyperparameters from Table 3 (note that the *optimal* hyperparameters were determined during hyperparameter tuning individually for each group of selected features). In general, as the number of features increased, the error trend was usually smoother when the modal hyperparameters were used.

## 4. Discussion

### 4.1. Model Comparison

Although all four supervised regression models (Lasso, SVR, random forest, and PLSR) had satisfactory prediction errors, errors from random forest were consistently higher (+10% average MAE across all input features). It was beyond the scope of this study to assess the computational efficiency of the models, but it may be worth noting that the SVR model took substantially more time to train than the other models. This was especially apparent during hyperparameter tuning, because the grid search method required that each model be trained for all possible hyperparameter combinations and all input features. If considering the computational requirement in addition to prediction accuracy, Lasso and PLSR generally emerged as the preferred models among those evaluated for predicting maize NUP using hyperspectral imagery.

There are many studies that correlate active or passive remote sensing with NUP in maize [14], but it seems rare that NUP predictions are evaluated with proper cross-validation techniques. The only report of cross-validated NUP prediction errors in maize was a study that used spectral indices from an active canopy sensor to predict NUP in the early vegetative development stages using simple linear regression [38]. Xia et al. [38] reported NUP RMSE as low as 16.6 kg ha$^{-1}$ across development stages (V5 to V10). The current study showed NUP RMSE as low as 13.6 kg ha$^{-1}$ across development stages (Figure 7), which was an 18% improvement compared to Xia et al. [38]. This improvement was observed despite the fact that predictions were made across a broader range of development stages

(V6 to V14) and NUP values (4.8 to 182 kg ha$^{-1}$) in the current study, for which larger errors can be expected due to increasing NUP variability for observations at later development stages.

Building cross-validated prediction models offers the advantage of interpreting remote sensing data in a way that is more extensible for practitioners. Rather than focusing on comparing interpretations of remote sensing information [14,39], remote sensing can instead be used to directly predict N status indicators familiar to agronomists (e.g., crop NUP, chlorophyll content, etc.). These N status indicators likely differ in their fundamental importance for making site-specific fertilizer recommendations, and thus the challenge of improving N recommendations can be broken down so research objectives focus instead on improving our fundamental understanding of N dynamics in maize (e.g., relating N status indicators to crop N demand). The value of comparing the accuracy and practicality of various remote sensing approaches [14] should not be diminished. Rather, we identify two independent requirements for making site-specific N recommendations using remote sensing: (i) the ability to reliably predict crop N status indicators familiar to agronomists, and (ii) having a clear understanding of how various N status indicators relate to soil N supply [12] and/or crop N requirement [13]. Thus, to improve the ability to provide reliable site-specific N fertilizer recommendations, we must not only be able to predict maize crop NUP, but we must understand how N supply and maize N requirement are expected to change throughout the season.

### 4.2. Segmentation

The goal for segmentation in this N study was to remove the pixels least likely to represent pure vegetation without having a bias against low chlorophyll plants. The MCARI2 spectral index was ideal for segmentation because it is sensitive to leaf area index and resistant to chlorophyll, which were important considering that plants with low N rates were expected to exhibit low chlorophyll. Shadowed pixels also tend to be problematic during segmentation, which MCARI2 handled especially well compared to other common spectral indices. Image pixel size slightly affected the range of MCARI2 histograms (Figure 2; a wider histogram corresponds to a higher coefficient of variation), even across the relatively small pixel sizes used in this study. The decreased variability observed for coarser pixel sizes tends to provide less specific biophysical crop information [40], but this issue was not severe for the small range in pixel sizes used in this study (i.e., 2.0–8.0 cm). Although early season weed control was not problematic at any of the sites used in this experiment, we do not expect basic segmentation methods (such as those used for this analysis) to discern differences between crop and weeds. Thus, from a practical perspective, the use of remote sensing for predicting biophysical parameters such as NUP requires suitable weed control, or perhaps a more sophisticated segmentation method that is capable of masking out weed pixels.

### 4.3. Inclusion of an Auxiliary Feature

A drawback of segmenting soil, shadow, and mixed pixels from the purest vegetation pixels is that information describing the canopy cover may be lost. It may be detrimental to segment generally unwanted pixels because their contribution may have a positive influence in explaining variability after averaging spectra from all pixels within a plot/region of interest and passing it to the regression model for training or validation. Therefore, it was desirable to identify a metric from the segmentation process that could be recorded and passed on to the subsequent steps for model evaluation so the benefits of segmentation could be exploited without sacrificing information about canopy cover. The 90th percentile MCARI2 was a suitable metric for segmenting pure vegetation, but it had a weak relationship with above-ground biomass due to saturation after the V10 development stage (Figure 4b). Since the 10th percentile MCARI2 did not saturate out as extensively (Figure 4a; i.e., there was a linear relationship across the range of biomass values), it was considered a more suitable metric for improving NUP prediction accuracy. This notion was supported by the improved overall prediction accuracy with the inclusion of the 10th percentile MCARI2 auxiliary feature. If segmentation of soil, shadow, or mixed pixels is possible (i.e., when using high-resolution aerial imagery), it is recommended to identify

a metric before the segmentation process that can be used as an auxiliary input feature, because it may improve overall model accuracy.

At any particular development stage, it was not surprising that the 90th percentile MCARI2 was lower for the larger pixel size—this simply illustrates that the larger pixel sizes are more influenced by the nearby soil and have a greater degree of mixing. The more interesting question is perhaps whether this greater degree of mixing observed with the coarser pixels translated to a bias and/or increased NUP prediction error from the models. Unfortunately, there was an offset in timing between image acquisition and tissue sampling for the Waseca whole field experiment at the V6 development stage, so prediction data are limiting for the 8.0 cm pixel size. However, with the data available, there is no evidence of a bias or increased prediction error from plots that had larger inherent pixel sizes. As long as the models are trained using data that are representative and within the constraints of data that will be used for predictions, these results show that imagery with up to 8.0 cm pixel size can be used during the early development stages (i.e., up to V14) without sacrificing NUP prediction accuracy.

### 4.4. Spectral Feature Selection

Hyperspectral sensors are incredible research tools because they offer hundreds of spectral bands, but the results of this study show that optimal prediction accuracy was usually achieved with only a few features. Features in the red edge (near 740 nm) and near-infrared (near 870 nm) regions were particularly important spectral regions for NUP (Figure 5). Reflectance in the red edge region changes rapidly due to the transition from strong pigment absorption in the red region to light scattering in leaves in the near-infrared region [41]. The Lasso algorithm oftentimes selected spectral features that were adjacent to each other (i.e., within approximately 6 nm). This not only illustrates the apparent importance of spectral precision and bandwidth of the sensor, but it also suggests that the slope in reflectance (i.e., derivative spectra) between adjacent bands is especially meaningful for explaining NUP. The inflection point in the red edge region is known to shift to shorter wavelengths as a consequence of reduced chlorophyll content [42], and perhaps the selection of adjacent spectral features is capturing this phenomenon.

### 4.5. Ongoing Challenges

The ability to predict spatially explicit NUP during the early- to mid-vegetative development stages has a clear value to maize producers that wish to make in-season N applications. Hyperspectral imagery is undoubtedly a promising tool for such an endeavor, but several technical and logistical challenges remain as described more fully below.

#### 4.5.1. Cost of Specialty Sensors

First, the cost of deploying an unmanned hyperspectral imaging unit is not likely to be practical for commodity crops such as maize. Until the cost of high-resolution, hyperspectral imaging substantially decreases, this technology is unlikely to be implemented. Only 2–5 features were typically required to achieve optimal or near-optimal NUP prediction performance regardless of the prediction model used in this study (Figure 8). Given the relatively few number of features needed for predicting NUP, it may be interesting for sensor engineers and manufacturers to explore the possibility of designing and building specialized narrowband sensors (i.e., up to five specific spectral bands with 2–3 nm bandwidth) that are useful for specific applications such as N management. Although hyperspectral imagery is a great research tool for identifying the most useful spectral bands, it certainly may not be required for satisfactory NUP prediction accuracy. Instead, more simple multispectral aerial sensors, satellite sensors, and active ground-based sensors may be sufficient for adequately predicting NUP. Considerable research has been conducted to predict early season maize NUP using these various sensors and/or platforms [14]. However, it is rare that prediction accuracy is reported using proper cross-validation techniques, making it challenging to compare among sensors, platforms, or prediction models to draw sound conclusions about the most satisfactory methods.

### 4.5.2. Timeliness

Second, it may be challenging sometimes to acquire, process, train, and analyze data in the time required to make a timely management recommendation. The burden of the acquisition process can be ameliorated with hardware that is fully integrated (e.g., spectral sensor that works well with an unmanned platform) and is efficient (e.g., battery capacity, flight time, read/write speeds, etc.). Pre-processing (e.g., radiometric calibration, reflectance conversion, georeferencing, etc.) and post-processing tasks can generally be improved with software, although customized solutions may have to be designed and built for specific applications and/or processing methodologies. Supervised models are only as good as their training and validation accuracies, and it would be wise to continually add new training data over time to incorporate data from different environments to improve the robustness of the models. While this study quantified the improvement in NUP prediction performance by including the 10th percentile MCARI2 as an input feature, it did not address the value of segmentation in the first place. This same notion could be said for many of the oftentimes subjective methods within the meticulous image processing workflow (e.g., spectral smoothing, spectral binning, segmentation, etc.). Furthermore, it is rare that a practitioner using the methods described in this study would be able to access the same resources used in this study (e.g., an imager with similar specifications and calibration protocol, processing software, etc.). For these reasons, more attention should be given toward quantifying the overall effect on prediction accuracy after mimicking the specifications of various popular image sensors or modifying thresholds used during processing (e.g., 90th percentile MCARI2 used as a segmentation threshold).

### 4.5.3. Making a Fertilizer Recommendation

Third, the scientific understanding of how early season crop NUP should be used to make a N fertilizer recommendation is poorly understood. Precision N management aims to match the N supply with crop demand in both space and time to ensure optimal grain yield while reducing the risk of environmental pollution from excess N. Thus, the prediction of early season crop NUP as demonstrated in this study is only useful for practical purposes when there exists a known relationship between observed NUP and future crop demand. Although there are cases when the relationship between plant-based spectral measurements and optimal N rate can be weak to non-existent [43], there is considerable evidence suggesting the contrary [44–48]. This indicates that remote sensing observations during the growing season can perhaps be used to determine crop N requirements, at least in part.

Future crop demand is itself subject to uncertainty because of the close relationship between the state of the soil environment (e.g., temperature, moisture, biological activity, etc.) and N additions (e.g., mineralization) or losses (e.g., leaching, denitrification, etc.). This is especially true in higher rainfall regions such as Minnesota and is largely driven by uncertainty in weather. Indeed, it is common that the economic optimum N rate (EONR) varies within a particular field and across years [10], but it is not always clear what drives these differences [39]. The lack of understanding of future crop demand together with uncertainty in the weather introduces a general distrust in any N recommendation that is derived from remote sensing [39]. Although there is always expected to be some degree of uncertainty in crop N demand, a planned in-season N fertilizer application can reduce the risk of N loss before the advent of increased demand and rapid crop uptake [39], ultimately improving NUP efficiency. However, the risk of an in-season application is that wet weather conditions can delay the planned application timing, which can subsequently lead to reduced grain yield and and/or increased residual soil N in some environments [49].

Future research should investigate the relationships between early season NUP and the EONR. Specifically, we must critically assess how spatially explicit, early season NUP is helpful for understanding the contributions to various parts of the N mass balance. Our understanding of net mineralization in soils is especially limiting [39], perhaps most fundamentally because of challenges with our ability to consistently estimate mineralizable N that varies due to soil properties, weather, sample collection timing, etc. [50]. However, there is perhaps an opportunity to use soil temperature and moisture sensors [51],

as well as other factors such as topography and soil physical and chemical properties, to estimate trends in mineralization somewhat reasonably throughout the season. The potential for incorporating process-based models to calculate EONR using readily available data also shows considerable promise [52] and should be explored for its ability to relate EONR with early season crop NUP.

## 5. Conclusions

As implemented in this study, supervised regression using hyperspectral remote sensing offers the opportunity to observe N supply spatially during the early- to mid-vegetative development stages, albeit in the form of crop N uptake rather than soil N supply. Introducing an auxiliary feature to the supervised regression models substantially improved N uptake prediction accuracy for all the models evaluated. Although hyperspectral imagery captured at a larger pixel size (i.e., up to 8.0 cm) demonstrated a greater degree of mixing (i.e., between soil, shadow, and vegetation), there did not appear to be evidence that this larger inherent pixel size translated to a bias or increased prediction error. The N uptake predictions from this study are superior to other reports in the literature that use various passive and active remote sensing techniques. As exciting and promising as this may be for making progress toward improved N management in agriculture, it has limited practical value for precision N management without additional efforts in three critical areas: (i) the cost of this specialty spectral data should be reduced, (ii) the timeliness from data acquisition to prediction should be improved, and (iii) the relationship between early season N uptake and upcoming crop N demand must be predicted with some degree of certainty.

To take the next step of making fertilizer recommendations, and to do so with greater confidence, it is imperative that remote sensing research in precision N management begins to encompass data-driven predictions of spatially explicit crop N demand and/or estimated optimum N rate. In continuing this journey, prediction accuracy must be reported habitually using proper cross-validation techniques, which is a practice that is rarely demonstrated in the relevant literature to date. Aside from preventing model overfitting, cross-validated results allow more authentic comparisons to be made among experiments that demonstrate the use of remote sensing for predicting N uptake and other biophysical crop parameters. Further research may conclude that narrowband or hyperspectral imagery is not a necessity for satisfactory prediction results (after all, optimal prediction accuracy was usually achieved with only a few spectral features). However, this must be properly quantified using various sensor configurations (e.g., spectral features, bandwidths, etc.), acquisition methods (e.g., altitude, platform, etc.), and/or processing methods.

**Author Contributions:** Conceptualization: T.J.N., C.Y., and D.J.M.; methodology: T.J.N. and G.D.P.; software: T.J.N.; formal analysis: T.J.N.; writing—original draft preparation: T.J.N.; writing—review and editing: C.Y., G.D.P., D.J.M., J.F.K., and F.G.F.; visualization: T.J.N.; project administration: D.J.M. and F.G.F.; funding acquisition: D.J.M., C.Y., T.J.N., and F.G.F. All authors have read and agreed to the published version of the manuscript.

**Funding:** This research was possible because of funding provided by the Minnesota Department of Agriculture through the Minnesota Clean Water Land and Legacy Act, grant number 153761 PO 3000031069 (Waseca experiments), and the Minnesota Soybean Research and Promotion Council, grant numbers 00079668 and 00071830 (Wells experiment). Minnesota's Discovery, Research, and InnoVation Economy (MnDRIVE) and the University of Minnesota also provided financial support in the form of student fellowships and/or research assistantships.

**Acknowledgments:** We thank Ali Moghimi for his contributions toward helping to integrate the hyperspectral imager with the UAV, as well as his valuable perspectives regarding image processing and model training. We thank the staff at the University of Minnesota Southern Research and Outreach Center, especially Jeff Vetsch, for help with field activities for the Waseca experiments. We also thank the Soil, Water, & Climate field crew, János Forgács, Xiaolei Deng, Abdullahi Abdullahi, and Leanna Leverich for assistance with sample collection, and Wenhao Su for assistance with image pre-processing.

**Conflicts of Interest:** The authors declare no conflict of interest. The funders had no role in the design of the study; in the collection, analyses, or interpretation of data; in the writing of the manuscript, or in the decision to publish the results.

## Appendix A

**Table A1.** Number of experimental observations (*n*), tissue sampling date, image acquisition date and local time, sampling area, number of tissue subsamples, and nitrogen extraction method for each experimental dataset subset by vegetative development stage.

| Year | Experiment | ID | Observation *n* | Stage | Sampling Date | Image Date | Image Time | Sample Area | Subsample *n* | Nitrogen Extraction |
|---|---|---|---|---|---|---|---|---|---|---|
| 2018 | Wells | 1 | 142 | V10 | 29 June 2018 | 28 June 2018 | 11:49–12:00 | 1.5 m × 5 m (2 rows) | 6 | Kjeldahl |
| 2019 | Waseca small-plot | 2 | 24 | V6 | 29 June 2019 | 29 June 2019 | 12:21–12:28 | 1.5 m × 2 m (2 rows) | 10 | Dry combustion |
| 2019 | | 3 | 24 | V8 | 9/10 July 2019 [1] | 09 July 2019 | 11:40–11:46 | 1.5 m × 2 m (2 rows) | 10 | Dry combustion |
| 2019 | | 4 | 24 | V14 | 23 July 2019 | 23 July 2019 | 12:03–12:09 | 1.5 m × 2 m (2 rows) | 6 | Dry combustion |
| 2019 | Waseca whole-field | 5 | 16 | V8 | 10 July 2019 | 08 July 2019 | 13:06–13:17 | 5 m × 10 m (6 rows) | 6 | Dry combustion |
| 2019 | | 6 | 16 | V14 | 23 July 2019 | 23 July 2019 | 12:32–12:42 | 5 m × 10 m (6 rows) | 6 | Dry combustion |

[1] V8 tissue sampling for the Waseca small-plot experiment began on 9 July and finished on 10 July.

**Table A2.** Relative mean absolute error (MAE) and relative root mean squared error (RMSE) for each of the models evaluated in this study. The relative error values in the *Spectral Features Only* column used five spectral features (739, 741, 801, 867, and 873 nm), and the relative error values in the *With Auxiliary Feature* column used four spectral features (737, 739, 867, and 871 nm) in addition to the 10th percentile MCARI2 (Modified Chlorophyll Absorption Ratio Index Improved) before image segmentation.

| Objective Function/Model | Spectral Features Only | With Auxiliary Feature |
|---|---|---|
| Relative [1] MAE | | |
| Lasso | 19.9% | 16.5% |
| Support vector | 18.8% | 17.0% |
| Random forest | 22.8% | 19.6% |
| Partial least squares | 19.8% | 16.7% |
| Relative [1] RMSE | | |
| Lasso | 26.8% | 23.8% |
| Support vector | 27.0% | 24.1% |
| Random forest | 31.2% | 27.4% |
| Partial least squares | 27.0% | 23.9% |

[1] The relative MAE and RMSE values were calculated by dividing the error value by the mean measured nitrogen uptake across all observations in the test set (57.1 kg ha$^{-1}$; *n* = 99).

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
