# Peer review of "Prediction of Early Season Nitrogen Uptake in Maize Using High-Resolution Aerial Hyperspectral Imagery"

_remotesensing, doi:10.3390/rs12081234_

Round 1

Reviewer 1 Report

Thank you for this interesting manuscript. I have several minor suggestions:

1) I suggest to better organize the results and report it by tables, which might be more informative, e.g. Figure 4 that I found to be very confusing 

2) The authors showed variability of the selected vegetation index, yet never referred to it, I suggest reading additional literature and add justification in the manuscript, e.g. 

Polinova, M., Jarmer, T. and Brook, A., 2018. Spectral data source effect on crop state estimation by vegetation indices. Environmental Earth Sciences77(22), p.752.

3) Some reported wavelengths are very close to each other, I found it meaningless and inapplicable. I strongly suggest testing models with reasonable spectral intervals. Moreover, to make it more applicable and interesting to the readers, I suggest testing models according to the available space-borne systems, taking into account new missions such as Venus and Proba. 

Reviewer 2 Report

Dear authors, 

Comments are added into the pdf file

Reviewer 3 Report

The article deals with evaluate learning techniques in estimating maize nitrogen uptake. The method is based on the use of spectral features carried out from high resolution aerial hyperspectral imagery. In order to reach the main objective of the study, captured images from airborne were used.

General comments

The topic of this article is relevant for the nitrogen fertilizer inputs. Because of the methodology presented, the study is also relevant for Remote Sensing Journal. The article is well written with the enough of details necessary to understand the approach. I would recommend this study for publication in Remote Sensing Journal, because it will be very useful for the scientific community about the methodology questions.

Specific comments

There are two small comments :

L83 : please explicit “MN”

L328 : close the parenthesis and end the sentence with a point.
